# A laboratory-based study to explore the use of honey-impregnated cards to detect chikungunya virus in mosquito saliva

Lisa Fourniol[1], Yoann Madec[2], Laurence Mousson[1], Marie Vazeille[1], Anna-Bella Failloux[1]*

1 Arboviruses and Insect Vectors Unit, Institut Pasteur, Paris, France, 2 Emerging Diseases Epidemiology Unit, Institut Pasteur, Paris, France

* anna-bella.failloux@pasteur.fr

**Data Availability Statement:** All relevant data are within the manuscript and its Supporting information files.

## Abstract

Mosquito control is implemented when arboviruses are detected in patients or in field-collected mosquitoes. However, mass screening of mosquitoes is usually laborious and expensive, requiring specialized expertise and equipment. Detection of virus in mosquito saliva using honey-impregnated filter papers seems to be a promising method as it is non-destructive and allows monitoring the viral excretion dynamics over time from the same mosquito. Here we test the use of filter papers to detect chikungunya virus in mosquito saliva in laboratory conditions, before proposing this method in large-scale mosquito surveillance programs. We found that 0.9 $cm^2$ cards impregnated with a 50% honey solution could replace the forced salivation technique as they offered a viral RNA detection until 7 days after oral infection of *Aedes aegypti* and *Aedes albopictus* mosquitoes with CHIKV.

## Introduction

Female mosquitoes are hematophagous arthropods that need blood for egg production and plant nectar as energy source for flying [1]. While blood is a key element for pathogen transmission, sugar source is pivotal for the survival of male mosquitoes and female fecundity [2]. Mosquitoes store sugar meals in large ventral diverticulum, the crop and sugar is poured out from time to time into the midgut for digestion and absorption [3]. When getting blood on a viremic host, a competent female mosquito ingests viral particles which penetrate inside midgut epithelial cells and replicate. Produced virions are released into the hemocele, where helped by the hemolymph, the virus reaches various internal organs including the salivary glands where it actively replicates. When the female mosquito has a subsequent blood meal, the virus is inoculated in the vertebrate host with the saliva delivered [4]. Once infected, the mosquito female remains infected for her entire life and able to transmit every time she bites [4]. Therefore, when feeding on a sugar source, saliva is excreted and if infectious, virus is expelled.

Mosquito-borne viruses have emerged during the past decades affecting millions of people [5]. Chikungunya hit a large belt of the tropical regions: islands of the Indian Ocean [6], Central Africa in 2007 [7], Americas in 2013 [8], and South Pacific region in 2014 [9].

**Funding:** This study was supported by the European Union's Horizon 2020 research and innovation program under EVAg grant agreement no. 653316 (https://www.european-virus-archive.com/) and the French Government's Investissement d'Avenir program, Laboratoire d'Excellence "Integrative Biology of Emerging Infectious Diseases" (grant n°ANR-10-LABX-62-IBEID). The funders had no role in study design, data collection and interpretation, or decision to submit the work for publication.

**Competing interests:** The authors have declared that no competing interests exist.

Chikungunya virus (CHIKV; *Alphavirus*, *Togaviridae*) is transmitted by the anthropophilic mosquitoes *Aedes aegypti* and *Aedes albopictus* which serve as vectors in an epidemic cycle [10]. CHIKV of the East-Central-South African genotype harboring an amino acid change at the position 226 (Ala to Val) in the envelope glycoprotein E1 is preferentially transmitted by *Ae. albopictus* and has become the most common CHIKV genotype worldwide [6, 11, 12].

Vector surveillance for tracking mosquito-borne pathogens is a tool for an early detection of arboviruses, and a help in designing appropriate intervention strategies prior to onset of human illness. Detection of arbovirus circulation is usually labor intensive and costly, imposing intensive captures of mosquitoes and mass screening for arboviruses [13]. The low prevalence of infection in mosquitoes [14] and the rapid degradation of viruses are the main issues that make this method irrelevant. Thus sampling mosquito saliva can be a gold standard if implemented properly [15]. Honey-coated cards have been used successfully to detect arbovirus circulation (namely, West Nile virus, Ross River virus, Barmah Forest virus, Japanese encephalitis virus and CHIKV) [16–19]. Here we explore the use of honey-impregnated cards for detecting CHIKV in laboratory conditions before proposing it as a method suitable for a surveillance system of arboviruses.

## Materials and methods

### Ethic statements

Animals were housed in the Institut Pasteur animal facilities (Paris) accredited by the French Ministry of Agriculture for performing experiments on live rodents. Work on animals was performed in compliance with French and European regulations on care and protection of laboratory animals (EC Directive 2010/63, French Law 2013–118, February 6th, 2013). All experiments were approved by the Institutional Animal Care and Use Committee (IACUC) at the Institut Pasteur (Ethics Committee #89 and registered under the reference APAFIS (Autorisation de Projet utilisant des Animaux à des FIns Scientifiques) #6573-201606l412077987 v2).

### Mosquito species tested

Two laboratory colonies were used: (i) *Ae. albopictus* Providence originally collected in 2010 in La Providence on La Réunion Island and maintained since then in insectaries; this population was involved in major outbreaks of CHIKV [6], and (ii) *Ae. aegypti* Paea collected in Tahiti (French Polynesia) and maintained in the laboratory since 1994 [20]. Mosquitoes were reared in standardized laboratory conditions. After egg hatching, larvae were distributed in pans of 200 individuals and supplied with a yeast tablet in 1 liter of dechlorinated water. Immature stages are maintained at $25\pm1°C$. Pupae were collected and placed in cages where adults emerged. Adults were fed with a 10% sucrose solution and kept at $28\pm1°C$ with a 16L:8D cycle and 80% relative humidity.

### Preparation of filter papers

To determine the effects of storage conditions on the ability of filter papers to preserve nucleic acids, 50 μL of serial viral dilutions (corresponding to 5 to $5\times10^6$ viral particles) were spotted on honey-coated $0.9\ cm^2$ heat-sterilized filter papers (Whatman, Piscataway, New Jersey). Honey (fir honey, France) solutions were proposed at 10%, 20%, or 50% prepared with distilled water. Cards of $0.9\ cm^2$ were incubated for up to 7 days at 28°C and 70% humidity. At day 1, 2, 3 and 7, exposed cards were immersed in 500 μL of FBS (fetal bovine serum) for 1h at +4°C and stored at -80°C until examination. Two cards (replicates) were prepared per viral dilution and honey concentration.

## Mosquito infections

Seven to ten day-old mosquito females were sorted at a rate of 60 individuals per box. For each species, five to six boxes were exposed to an infectious blood meal containing 1.4 mL of washed rabbit erythrocytes, 700 μL of CHIKV suspension and ATP at 1 mM as a phagostimulant. CHIKV strain (06.21) isolated from a patient on La Réunion Island in 2005 [6], belonging to the East-Central-South African genotype, was kindly provided by the French National Reference Center for Arboviruses; viral stocks were produced after two passages on C6/36 cells in T75 flasks. Briefly, subconfluent C6/36 cells were inoculated with 500 μL of viral suspension at a 0.1 MOI (multiplicity of infection) and incubated at 28˚C. After 1 h of adsorption, 10 mL of Leibovitz's L-15 medium (Life Technologies) complemented with 2% fetal bovine serum (FBS; Thermo Fisher Scientific), 1X of non-essential amino acids (NAA; Life Technologies), 1X of Penicillin-Streptomycin (Life Technologies) were added to the flask. After 2 days of incubation, the cell culture supernatant was harvested and the percentage of FBS adjusted to 10%. Aliquots were stored at -80˚C until used. The titer of the blood meal was at $10^{6.5}$ plaque-forming unit (pfu)/mL. Four Hemotek® (Hemotek Ltd, Blackburn, UK) feeders containing the infectious blood meal were prepared and each box was placed under a feeder for 15 min. Then mosquitoes fed through a piece of pork intestine covering the base of a feeder maintained at 37˚C. After feeding, fully fed mosquitoes were isolated in cardboard containers and maintained with 10% sucrose under controlled conditions (28±1˚C, relative humidity of 80%, 16h light:8 h dark cycle) for up to 7 days. After feeding, mosquitoes were individually isolated in 50 mL Falcon® tube closed with a mesh on the top. Females had then access to a 0.9 cm$^2$ card impregnated with a 50% honey solution put on the mesh. Mosquitoes were maintained at 28˚C and 70% humidity. To define whether the quantity of virus detected depends on contact time between cards and mosquitoes, cards were unchanged until day of examination or changed one day before examination. At 3 and 7 days post-exposure (dpe), honey-impregnated cards were prepared for RNA extraction and viral quantification. To compare with the standard technique of forced salivation [21], mosquitoes were examined at 3 and 7 days after infection; wings and legs were removed from each mosquito, and the proboscis was inserted into a 20 μL tip containing 5 μL FBS. After 30 min, saliva-containing FBS was expelled in 45 μL of Leibovitz L-15 medium (Life Technologies) for quantification. Three experimental infections (replicates) were performed per condition. Non-infected mosquitoes (exposed to non-infectious blood or culture media) were not tested.

## Nucleic acid extraction and quantitative RT-PCR

Cards were immersed in 500 μL of Leibovitz L15 medium (Invitrogen, CA, USA) supplemented with 10% fetal bovine serum (FBS) for 1h at +4˚C. After homogenization and centrifugation, the supernatant containing viral particles was stored at -80˚C until use. Samples were processed for RNA extraction using the Nucleospin® RNA II kit (Macherey-Nagel, Hoerdt, France) followed by one-step RT-PCR performed in a volume of 25 μL containing 3 μL RNA template, 12.5 μL 2X Brilliant SYBR Green I QPCR Master Mix (Stratagene), 1 μL sense (2.5 μM), 1 μL anti-sense (2.5 μM), 0.25 μL Fluorescein (1 μM), and 0.0625 μL Stratascript RT/ RNAse block enzyme. Primers were selected in the E2 structural protein coding region [11]: sense Chik/E2/9018/+ (`CACCGCCGCAACTACCG`) and anti-sense Chik/E2/9235/- (`GATTGGTGACCGCGGCA`). The amplification program in a CFX96 Real-Time System (Biorad, CA) included: a reverse transcription at 50˚C for 10 min, a step of reverse transcriptase inactivation at 95˚C for 1 min followed by 40 cycles of 95˚C 10 s and 60˚C 30 s (annealing-extension step). After amplification, a melting curve was acquired to check the specificity of PCR products. PCR was performed in duplicate for each sample. Signals were normalized

to the standard curve using 10-fold serial dilutions of viral RNA ($10^1$ to $10^8$). Normalized data were used to measure the number of RNA copies according to the $\Delta C_t$ analysis. The number of RNA copies detected corresponded to the quantity of viral particles tested (S1 Table).

### Statistical analysis

To evaluate the ability of honey-impregnated cards to preserve nucleic acids, we first compared the proportion of samples with detected virus according to virus dilution, honey concentration and day post-infection (dpi) using chi-square tests. In a second step, in samples with virus detected, we compared RNA copies detected according to virus dilution, honey concentration and dpi using Kruskall-Wallis non-parametric tests. A linear regression model was used to evaluate the adjusted effects of virus dilutions, honey concentration, and dpi.

In experiments with mosquitoes, proportions of females showing transmission were compared between species, dpe and treatments using logistic regression models. Then, in mosquitoes showing transmission, the level of viral RNA detected was compared between species and dpe using Student's t test, and between treatments using analysis of variance (ANOVA).

Statistical analyses were conducted using the Stata software (StataCorp LP, Texas, USA). p-values < 0.05 were considered significant. If necessary, the significance level of each test was adjusted based on the number of tests run, according to the sequential method of Bonferroni.

## Results

### Viral RNA detection on filter papers

Before conducting sugar-feeding experiments with mosquitoes, we determined the effects of several conditions on the ability of filter papers to preserve nucleic acids when impregnated with honey solutions provided at three different concentrations (10%, 20%, and 50%). We found that the proportion of honey-impregnated cards allowing virus detection did not vary according to the day post-infection (dpi) (chi-square test: p = 0.08) or honey concentration (10%, 20% or 50%) (chi-square test: p = 0.47). Fig 1a–1c showed the quantity of viral RNA detected on cards impregnated with different dilutions of honey (10%, 20%, 50%) and on which, were deposited different quantities of viral particles (5 to $5x10^6$ pfu) (details in S2 Table). When only considering the samples with virus detected, the number of viral RNA detected on cards did not vary according to dpi (Kruskal-Wallis test: p = 0.10) but increased as the honey concentration increased (Kruskal-Wallis test: p = 0.037). Moreover, the number of viral RNA detected on cards decreased as the virus dilution increased (Kruskal-Wallis test: p <0.001). Using a linear regression and adjusting to the quantity of viral particles deposited and the dpi, we showed a significant effect of the honey concentration with a higher viral RNA detected on cards impregnated with a 50% honey dilution (logistic regression model: p < 0.001). One viral CHIKV RNA can be detected 7 days after depositing one viral particle on filter papers impregnated with a 50% honey solution (Fig 1c). Altogether, the number of viral RNA detected on filter papers did not vary according to the dpi but vary with the honey dilution; the 50% dilution of honey preserved better viral RNA on cards.

### Detection of viral RNA on honey-impregnated filter papers placed in contact with mosquitoes orally infected with a CHIKV-infectious blood meal

To define whether filter papers can replace saliva collection by forced salivation, *Ae. aegypti* and *Ae. albopictus* previously exposed to an infectious blood meal were put individually in

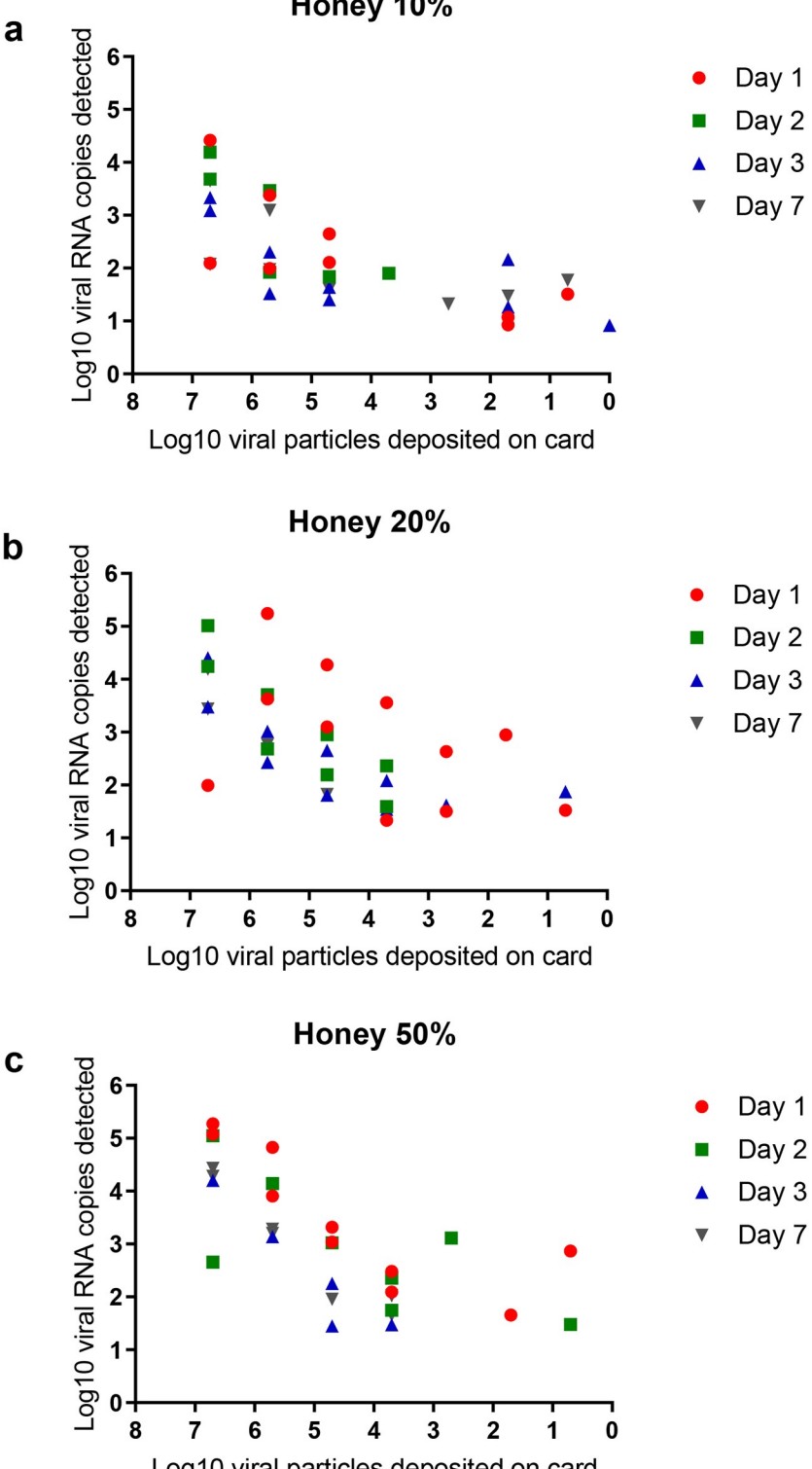

**Fig 1. Viral RNA copies detected on 0.9 cm$^2$ spotted with different quantities of CHIKV particles (5 to 5x10$^6$ pfu) and impregnated with honey solutions (10%, 20%, 50%), examined at day 1, 2, 3 and 7.** Different quantities of viral particles were deposited on 0.9 cm$^2$ cards imbided with honey solutions. After 1, 2, 3 and 7 days of incubation at 28˚C, cards were immersed in 500 μL of FBS (fetal bovine serum) for 1h at +4˚C. Samples were processed for RNA extraction and qRT-PCR. Two replicates were performed. Each dot represents one card; detailed information in S2 Table.

**Table 1. Transmission efficiencies (%) of *Aedes aegypti* Paea and *Aedes albopictus* La Providence, 3 and 7 days after exposure to an infectious blood meal containing $10^{6.5}$ pfu/mL of CHIKV.** Saliva were collected using the forced salivation technique and quantified by RT-qPCR. In brackets, number of mosquitoes tested.

| | Species | *Aedes aegypti* Paea | | | *Aedes albopictus* La Providence | | |
|---|---|---|---|---|---|---|---|
| | Replicate | 1 | 2 | 3 | 1 | 2 | 3 |
| Card unchanged | *Day3* | 30% (10) | 55.55% (9) | 27.27% (11) | 0% (5) | 50% (10) | 71.42% (14) |
| | Mean (N) | 36.66% (30) | | | 51.72% (29) | | |
| | *Day 7* | 50% (10) | 50% (8) | 80% (10) | 60% (5) | 75% (4) | 42.85% (14) |
| | Mean (N) | 60.71% (28) | | | 52.17% (23) | | |
| Card changed | *Day3* | 30% (10) | 30% (10) | 54.54% (11) | 20% (5) | 62.5% (8) | 64.28% (14) |
| | Mean (N) | 38.70% (31) | | | 55.55% (27) | | |
| | *Day 7* | 60% (10) | 80% (5) | 36.36% (11) | 40% (5) | 0% (2) | 37.5% (8) |
| | Mean (N) | 53.84% (26) | | | 33.33% (15) | | |
| Saliva collection | *Day3* | 35% (40) | 35% (20) | 55% (20) | - (0) | 30% (20) | 60% (20) |
| | Mean (N) | 40% (80) | | | 45.0% (40) | | |
| | *Day 7* | 96.42% (28) | 80% (20) | 85% (20) | 75% (8) | 100% (20) | 100% (20) |
| | Mean (N) | 88.23% (68) | | | 95.83% (48) | | |

contact with filter papers prepared under two conditions (status unchanged and changed) and examined at 3 and 7 dpe; the status "unchanged" refers to filter papers that were placed on the top of the tube and kept as it is until the day of examination and the status "changed" to filter papers placed on the top of the tube that were changed one day before the examination. Filter papers were all impregnated with a 50% honey solution. When examining the transmission efficiencies (TE, corresponding to the proportion of mosquitoes with infectious saliva among mosquitoes tested), the three replicates were similar (Fisher's exact test, p > 0.05) allowing to pool all individuals exposed to the same treatment (Table 1).

A total of 257 mosquitoes (among 445; 57.7%) were able to expectorate the virus. Dpe (3, 7) ($p < 10^{-4}$) and treatments (card unchanged, card changed and saliva collection) (p = 0.0007) affected significantly TE while mosquito species (*Ae. aegypti*, *Ae. albopictus*) did not (p = 0.25) (S3 Table). After adjusting to dpe, treatments remained significantly correlated with TE ($p < 10^{-4}$) and more specifically, for saliva collection (p = 0.001). When examining the number of viral RNA detected, loads were significantly higher at 7 dpe compared to 3 dpe (Student's test: $p < 10^{-3}$) and did not vary according to the mosquito species (Student's test: p = 0.42) and treatments (ANOVA: p = 0.75).

For *Ae. aegypti* mosquitoes, TEs calculated using filter papers did not increase with the dpe (3 and 7) whatever the status of filter papers (unchanged versus changed) (Fisher's exact test, p > 0.05). However, saliva collection by forced salivation allowed detecting more mosquitoes with infectious saliva at 7 dpe (Fisher's exact test, $p < 10^{-4}$).

For *Ae. albopictus*, TEs did not increase with the dpe (Fisher's exact test, p > 0.05) and as for *Ae. aegypti*, we detected more mosquitoes with infectious saliva by forced salivation (Fisher's exact test, $p < 10^{-4}$). Finally, *Ae. aegypti* and *Ae. albopictus* behaved similarly whatever the dpe and the way to detect saliva-excreted viruses (Fisher's exact test, p > 0.05).

When analyzing the number of viral RNA expectorated by mosquitoes (Fig 2), neither the species, the dpi, nor the status of filter papers (unchanged and changed) affected significantly the quantities of viral RNA detected (Kruskal-Wallis test: p > 0.05) with only one exception, the number of viral RNA being higher in changed filter papers at 7 dpe for *Ae. aegypti* (Kruskal-Wallis test: p = 0.009).

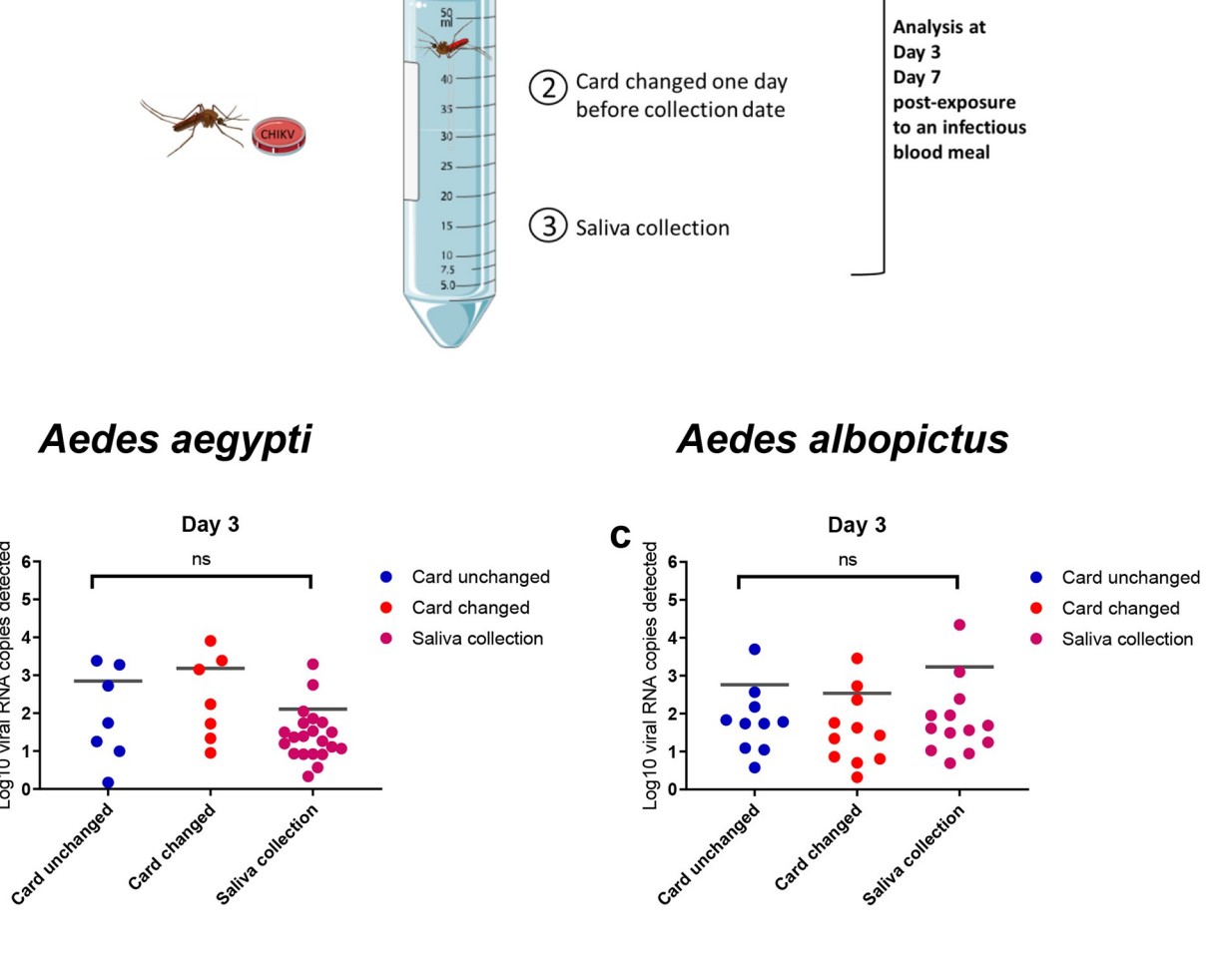

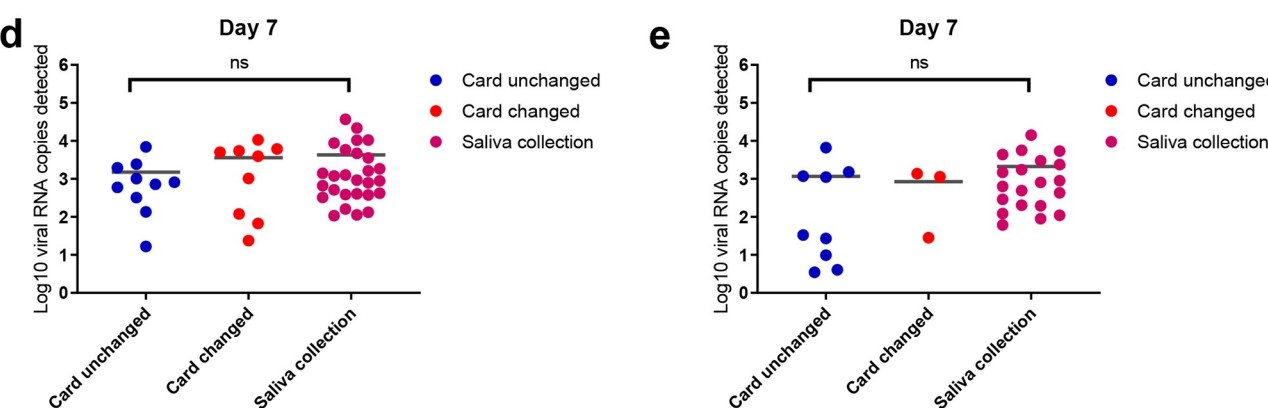

**Fig 2.** CHIKV-infected saliva detected on cards, 3 and 7 days after exposure of *Aedes aegypti* (b, d) and *Aedes albopictus* (c, e) to an infectious blood meal. (a) Mosquitoes were exposed to a CHIKV infectious blood meal at $10^{6.5}$ pfu/mL and maintained in individual tubes at 28°C. A 0.9 $cm^2$ of 50% honey impregnated card was deposited on the top of tubes. Filter papers were (1) changed one day before examination or (2) kept unchanged until examination, and compared to the typical (3) saliva collection using the forced salivation technique. The number of viral RNA copies were estimated by qRT-PCR. Each dot represents an individual mosquito. Between 11 to 60 mosquitoes were analyzed for *Ae. aegypti* and 5 to 46 for *Ae. albopictus*. Three replicates were performed per condition. Bars indicate the mean. ns (non-significant) indicates the lack of statistical significance (p > 0.05).

## Discussion

Our study shows that filter papers could replace the forced salivation technique to detect viral RNA in saliva of *Ae. aegypti* and *Ae. albopictus*. CHIKV RNA can be detected in saliva deposited on filter papers until 7 days after oral infection of mosquitoes. Moreover, viral CHIKV RNA can be detected 7 days after being deposited on filter papers impregnated with a 50% honey solution. To summarize, viral RNA can be detected on filter papers 14 days after oral infection of mosquitoes.

Our pilot experiment shows that the filter papers impregnated with a honey solution can preserve viral RNA until day 7 post infection. We are able to detect one viral RNA copy on filter papers coated with a 50% honey solution. This method has been used successfully to detect different arboviruses (e.g. West Nile virus (WNV), Ross River virus (RRV), Barmah Forest virus (BFV), Japanese encephalitis virus (JEV), and CHIKV) [16–19]. Its gives results comparable to those obtained using the standard forced salivation technique of mosquitoes [16]. An early warning of virus circulation prior detection in humans can be performed using immunologically naïve sentinel animals [22] and field-collected mosquitoes [13]. While sentinel animals raise some ethical concerns and restrictive constraints (methodological, financial), tracking arboviruses in mosquitoes have more advantages. However, mass screening of mosquitoes for arboviruses requires collecting thousands of mosquitoes that should be stored in appropriate conditions to preserve viral RNA and the infection rate of mosquitoes is usually low [13]. The use of filter papers to collect mosquito saliva exploits the biological need of mosquitoes to feed on a sugar source [16]; infectious mosquitoes will be brought to excrete the virus on filters papers during sugar feeding. This method being non-destructive, monitoring the viral excretion dynamics over time is made possible from the same mosquito. Using these papers in combination with trapping mosquitoes (e.g. using the BG-Sentinel traps) can be a promising tool for detecting mosquito-borne viruses. A 50% honey solution promotes viral RNA stability for up to 7 days, compatible with a surveillance system applicable on the field. One disadvantage of using filter papers is that the identity of the mosquito that has expectorated the virus, i.e. the potential vector cannot be determined.

Viral CHIKV RNAs are successfully detected from day 3 post-exposure with comparable quantities, whether the excreted virus is deposited on filter papers or excreted from mosquitoes. Successful viral transmission by a mosquito will occur if the virus overcomes at least two different anatomical barriers: the midgut and the salivary glands [23]. The midgut is the first barrier where the virus cannot penetrate inside the epithelial cell after being ingested with the infectious blood meal (i.e. midgut infection barrier) and/or the virus cannot escape into the hemocele after replication in epithelial cells, infecting different internal organs including the salivary glands (i.e. midgut escape barrier). The salivary glands correspond to the second barrier with also a salivary gland infection barrier and a salivary gland escape barrier. When the virus is detected in the expectorated mosquito saliva, it means that the mosquito is capable of transmission and the time interval between virus acquisition during blood feeding and the transmission is referred to as the extrinsic incubation period (EIP); EIP is the most critical parameter for virus transmission and will condition the choice of the vector control strategy [4]. Indeed, vector control will aim in reducing the mosquito lifespan to decrease the probability of successful transmission (e.g. for *Ae. aegypti* and *Ae. albopictus*, mean EIP for CHIKV is 2–3 days at 28˚C [21], imposing to implement vector control measures promptly after human cases are detected). However, the forced salivation technique does not allow determining the physiological dose of virus delivered by mosquitoes when bite [21]. This method allows measuring accurately the vector competence of a mosquito species that must be easily reared and infected in laboratory conditions. These constraints exclude to study a number of mosquitoes not adapted to laboratory conditions.

In conclusion, we show that viral CHIKV RNA can be detected in mosquito saliva deposited on filter papers soaked with a honey solution. This method can advantageously replace the forced salivation technique for assessing vector competence in laboratory [24]. The use of honey-impregnated filter in field conditions may help to better monitor the risk of transmission and anticipate the emergence.

## Supporting information

**S1 Table. Detection sensitivity of CHIKV RNA copies by qRT-PCR.** 50 μL of different dilutions of viral particles were processed for RNA extraction and quantification of RNA copies by qRT-PCR.
(PDF)

**S2 Table. Quantities of RNA copies detected on cards impregnated with honey solutions (10%, 20%, 50%) and examined at different days (1, 2, 3 and 7) after spotting different quantities of CHIKV particles.** Two replicates (R1 and R2) were performed.
(PDF)

**S3 Table. Comparison of transmission efficiencies between mosquito species, day post-exposure and treatments (logistic regression model).**
(PDF)

## Acknowledgments

We thank Malika Hocine and Jocelyne Alexandre for helping in rearing mosquitoes.

## Author Contributions

**Conceptualization:** Anna-Bella Failloux.

**Formal analysis:** Yoann Madec.

**Funding acquisition:** Anna-Bella Failloux.

**Investigation:** Lisa Fourniol.

**Methodology:** Laurence Mousson, Marie Vazeille.

**Supervision:** Anna-Bella Failloux.

**Writing – original draft:** Anna-Bella Failloux.

**Writing – review & editing:** Yoann Madec, Marie Vazeille.

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
