## [Decision Letter · Decision Letter 0]

8 Feb 2021

PONE-D-20-36874

A laboratory-based study to explore the use of honey‑impregnated cards to detect chikungunya virus in mosquito saliva

PLOS ONE

Dear Dr. Failloux,

Thank you for submitting your manuscript to PLOS ONE. After careful consideration, we feel that it has merit but does not fully meet PLOS ONE’s publication criteria as it currently stands. Therefore, we invite you to submit a revised version of the manuscript that addresses the points raised during the review process. 

Please try to carefully study and respond all the queries raised by both reviewers before you attempt to return your revised manuscript for further revision.

Please submit your revised manuscript by March, 15th. If you will need more time than this to complete your revisions, please reply to this message or contact the journal office at plosone@plos.org. Please include the following items when submitting your revised manuscript:

We look forward to receiving your revised manuscript.

Kind regards,

Luciano Andrade Moreira, PhD

Academic Editor

PLOS ONE

Journal Requirements:

2)  Thank you for including your ethics statement:  "Animals were housed in the Institut Pasteur animal facilities (Paris) accredited by the French Ministry of Agriculture for performing experiments on live rodents. Work on animals was performed in compliance with French and European regulations on care and protection of laboratory animals (EC Directive 2010/63, French Law 2013-118, February 6th, 2013). All experiments were approved by the Ethics Committee #89 and registered under the reference APAFIS#6573-201606l412077987 v2.".   

Please amend your current ethics statement to include the full name of the ethics committee that approved your specific study.

For additional information about PLOS ONE submissions requirements for ethics oversight of animal work, please refer to http://journals.plos.org/plosone/s/submission-guidelines#loc-animal-research  

Reviewers' comments:

Reviewer's Responses to Questions

**Comments to the Author**

1. Is the manuscript technically sound, and do the data support the conclusions?

Reviewer #1: Partly

Reviewer #2: Partly

2. Has the statistical analysis been performed appropriately and rigorously? 

Reviewer #1: Yes

Reviewer #2: Yes

3. Have the authors made all data underlying the findings in their manuscript fully available?

Reviewer #1: Yes

Reviewer #2: Yes

4. Is the manuscript presented in an intelligible fashion and written in standard English?

Reviewer #1: Yes

Reviewer #2: Yes

5. Review Comments to the Author

Reviewer #1: In their manuscript ‘A laboratory-based study to explore the use of honey‑impregnated cards to detect chikungunya virus in mosquito saliva’, the authors examine the potential use of honey-infused filter papers as a diagnostic tool for CHIKV infection in Aedes aegypti and Aedes albopictus mosquitoes using experimental infection and salivation assays.

They determined that cards infused with 10%, 20% and 50% honey solutions are suitable for virus detection out to 7 days post-expectoration, and that the 50% treatment was most effective at preserving viral RNA. They then compare transmission efficiencies for both species, for cards against the traditional forced salivation assay and show that both approaches could be utilized, although a significantly greater proportion of positive samples were detected with the forced salivation approach. The study is a useful examination of a tool that could prove to be an important form of arboviral diagnostic in the future. However, in its current form, the manuscript is somewhat disorganized and key information necessary for readers to fully interpret the results, or repeat the assays is missing. Please see specific comments below:

1. Lines 51- 53 “CHIKV of the East-Central-South African genotype harboring an amino acid change at the position 226 (Ala to Val) in the envelope glycoprotein E1 is preferentially transmitted by Ae. albopictus (6, 11, 12).” – Given the inclusion of this statement, the authors should state whether the virus isolate used in the study has this genotype, and consider what that implies for their results.

2. Details that should be added to/clarified in the methods section:

• Methods – Were any uninfected controls to help gauge the false positive rate?

• Were the filter papers sterilized prior to experiments?

• How many replicates were performed for each experiment? There is some conflicting information. “Five to six batches of 60 females” – Does this mean five independent experimental infections were performed, or were all mosquitoes fed at one time but in five different containers?

• “CHIKV strain (06.21) was isolated from a patient on La Réunion Island in 2005 (6);viral stocks were produced after two passages on 101 C6/36 cells” – Does this mean that the virus was only ever passaged twice over 15 years? More specific details about viral isolate history, and pre-experimental passaging/storage are needed.

• Lines 111-112 - “with the standard technique of forced salivation (21), wings and legs were removed from each mosquito, and the proboscis was inserted into a 20 μL tip containing 5 μL FBS. – The phrasing is a little confusing, as it sounds like wings/legs were removed for mosquitoes feeding on the filter paper too. Should state the time points post-exposure when salivation experiments were performed.

3. The introduction would be better structured if the first and second paragraphs were swapped.

4. Lines 60-62 “Honey-coated cards have been used successfully to detect arbovirus circulation (namely, West Nile virus, Ross River virus, Barmah Forest virus, Japanese encephalitis virus and CHIKV) (16-19).” – what is different between this historical CHIKV study and the current study?

5. Cards were homogenized in 500uL of medium. Saliva samples in 5uL FBS were added to 45uL medium. This is a 10x difference in media volume, and could have led to differences in detection rates between the two approaches, depending on their relative sensitivity. Was this accounted for in downstream calculations and statistical comparisons.

6. The terms days post-exposure and days post-infection are both used throughout the manuscript. For the sake of consistency, only one should be used.

7. The Fig 2 legend does not do a good job of describing the figure and should be re-written. From the data in these panels, some treatment groups have a smaller sample size than anticipated based on what is described in the methods (one has an N of 3). The sample sizes for individual treatment groups should be described somewhere. Legends should also state whether data were pooled across replicates, or if they represent a single experiment.

8. The discussion section could be improved on in the following ways:

• The section overlooks the authors results which demonstrate that filter cards are less accurate at detecting virus in saliva than the forced salivation method. An objective discussion of the strengths and weaknesses of the two approaches is important.

• The EIP discussion from line 252 is very general and does not provide supporting evidence to the authors argument that honey-impregnated filter cards are a useful tool for arbovirus surveillance.

• Some key advantages of the filter cards are overlooked: they could be deployed in the field, left for a week and collected – a feeding station for surveillance, similar to sentinel birds for WNV. This would offer a direct measurement not feasible for other approaches. Honey promotes RNA stability, overcoming a major weakness with other detection methods.

9. Abstract – “Detection of virus in mosquito saliva using honey-impregnated filter papers seems to be a promising method.” – This sentence is vague. Why, specifically, is this a promising method?

10. Lines 17-18 “Mosquito control is implemented when arboviruses are detected in patients or in field collected mosquitoes.” – It is more commonly implemented pre-emptively in areas of known transmission.

11. Lines 162-163 - “and as the virus dilution increased (Kruskal-Wallis test: p <0.001).” – This reads as though more virus was detected when samples were more diluted, but the figure shows the opposite effect.

12. Line 169 – “50% preserving better viral RNA.” This is confusing and needs to be rephrased.

13. Lines 178-179 – “Detection of viral RNA on honey-impregnated filter papers proposed to CHIKV orally-infected mosquitoes” – This is confusing and needs to be rephrased.

14. Lines 180-183 “To define whether filter papers can replace saliva collection by forced salivation, Ae. aegypti and Ae. albopictus previously exposed to an infectious blood meal were put individually in contact with filter papers prepared under two conditions (status unchanged and changed) and examined at 3 and 7 dpe.” – This is a little difficult to understand, and it would be helpful if definitions for ‘changed’ and ‘unchanged’ were included in the text. In the legend for figure 2, the process is not well defined.

15. Line 212 “viral RNA excreted” – should read viral RNA expectorated

16. Lines 248-249 “This method has the advantage of being non-destructive as is the technique of forced salivation” – I think the phrasing here needs a little work as it makes it sound as though you are arguing that forced salivation is not destructive, even though it typically involves removing the legs and wings of the mosquitoes involved.

17. Table 1 – mean transmission efficiency units (% I assume) should be included in the table.

Reviewer #2: Mosquito-borne viruses, including Chikungunya virus (CHIKV), represent a permanent and increasing global public health threat. Prevention and control strategies for arboviral diseases include continuous surveillance of arboviruses transmitting mosquito populations. However, systematic detection of infected vectors on the field remains fastidious. In this work, Fourniol et al., assessed the use of honey impregned cards as a laboratory tool to routinely and efficiently detect CHIKV in mosquito saliva. By deposing controlled amounts of viruses on honey impregned cards, the authors determined an optimal honey concentration for a better recovery and conservation of viral RNA. They finally demonstrated that detection of viral RNA in mosquito saliva using honey cards was as efficient as the use of the well-established forced saliva method. Taken together, these results are of interest for field application in monitoring the risk of arboviruses transmission and may also inspire the development of similar assays in other vector insects transmitting different pathogens. While this work is of interest for the field, the paper would benefit of language improvement, proofreading, and authors should address the following questions/concerns:

1) The authors should clarify how their work is novel compared to the already published study that was using the same approach to detect CHIKV RNA in mosquito saliva (doi: 10.1073/pnas.1002040107).

2) As the transmission efficiency is mosquito dependent, it is rather complicated to compare the results obtained with the honey cards and the forced salivation without taking this parameter into account. Using forced salivation on the mosquitoes that were exposed to the honey cards to detect viral RNA would have allowed a better comparison between the different detection methods.

3) The authors should explain the aim of testing the honey cards in two distinct conditions (changed and unchanged).

4) Related to the Table 1: The authors indicated that the replicates obtained for each condition were similar (line 185). Could the authors confirm this statement as the variations for A. albopictus at Day 3 vary from 0 to 71 (card unchanged) and from 20 to 64.28 (card changed).

5) The titer of the virus is indicated as 106.5 pfu/mL. This annotation is quite unusual, could the authors confirm that they mean 3.16 x 106 pfu/mL?

6) The authors should provide, to the best extent possible, all the information in the figure legend and make table clearer so the reader can understand the figures:

a. In Figure 1, it is not clear what each dot represents. Indeed, if one dot represents one honey card then the authors should make visible on the graph all the data points, even when no viral RNA copies were detected. Another alternative would be to indicate above each virus dilution how many honey cards were tested.

b. In Table 1, it is not clear what the unbracketed number refer to.

c. In Table 2, it is not clear how the treatments were spread across the two different mosquito species.

d. In Table 2, it is difficult to assess what data comparison the p-values are referring to.

e. In Figure 2, statistic could be indicated even if p-values are not significant.

7) Please refer to the figures in the text when necessary.

a. No mention of Figure 1a, 1b.

b. Text in lane 199 and between lane 199-203 should refer to table S2.

6. PLOS authors have the option to publish the peer review history of their article (what does this mean?). If published, this will include your full peer review and any attached files.

Reviewer #1: **Yes: **Eric P. Caragata

Reviewer #2: No

---

## [Author Response · Author response to Decision Letter 0]

3 Mar 2021

Answers to Reviewer #1

1) Lines 51- 53 “CHIKV of the East-Central-South African genotype harboring an amino acid change at the position 226 (Ala to Val) in the envelope glycoprotein E1 is preferentially transmitted by Ae. albopictus (6, 11, 12).” – Given the inclusion of this statement, the authors should state whether the virus isolate used in the study has this genotype, and consider what that implies for their results.

We understand and have added: “and has become the most common CHIKV genotype worldwide” in line 52, and “CHIKV strain (06.21) isolated from a patient on La Réunion Island in 2005 (6), belonging to the East-Central-South African genotype, was kindly provided by the French National Reference Center for Arboviruses;” in lines 100-102.

2) Details that should be added to/clarified in the methods section:

• Methods – Were any uninfected controls to help gauge the false positive rate?

To answer properly to this question, we need more details.

• Were the filter papers sterilized prior to experiments?

Filters have been sterilized before use. We have added in lines 89 “heat-sterilized filter papers”

• How many replicates were performed for each experiment? There is some conflicting information. “Five to six batches of 60 females” – Does this mean five independent experimental infections were performed, or were all mosquitoes fed at one time but in five different containers?

For more consistency, this section has been rearranged from line 97 to line 113: “Seven to ten day-old mosquito females were sorted at a rate of 60 individuals per box. For each species, five to six boxes were exposed to an infectious blood meal containing 1.4 mL of washed rabbit erythrocytes, 700 µL of CHIKV suspension and ATP at 1 mM as a phagostimulant. CHIKV strain (06.21) isolated from a patient on La Réunion Island in 2005 (6), belonging to the East-Central-South African genotype, was kindly provided by the French National Reference Center for Arboviruses; viral stocks were produced after two passages on C6/36 cells in T75 flasks. Briefly, subconfluent C6/36 cells were inoculated with 500 µL of viral suspension at a 0.1 MOI (multiplicity of infection) and incubated at 28°C. After 1 h of adsorption, 10 mL of Leibovitz’s L-15 medium (Life Technologies) complemented with 2% fetal bovine serum (FBS; Thermo Fisher Scientific), 1X of non-essential amino acids (NAA; Life Technologies), 1X of Penicillin-Streptomycin (Life Technologies) were added to the flask. After 2 days of incubation, the cell culture supernatant was harvested and the percentage of FBS adjusted to 10%. Aliquots were stored at -80°C until used. The titer of the blood meal was at 106.5 plaque-forming unit (pfu)/mL. Four Hemotek® (Hemotek Ltd, Blackburn, UK) feeders containing the infectious blood meal were prepared and each box was placed under a feeder for 15 min. Then mosquitoes fed through a piece of pork intestine covering the base of a feeder maintained at 37°C.”

• “CHIKV strain (06.21) was isolated from a patient on La Réunion Island in 2005 (6);viral stocks were produced after two passages on C6/36 cells” – Does this mean that the virus was only ever passaged twice over 15 years? More specific details about viral isolate history, and pre-experimental passaging/storage are needed.

The viral stock was constituted once and stored in aliquots until use. Details on the virus production has been added from line 102 to line 109: “viral stocks were produced after two passages on C6/36 cells in T75 flasks. Briefly, subconfluent C6/36 cells were inoculated with 500 µL of viral suspension at a 0.1 MOI (multiplicity of infection) and incubated at 28°C. After 1 h of adsorption, 10 mL of Leibovitz’s L-15 medium (Life Technologies) complemented with 2% fetal bovine serum (FBS; Thermo Fisher Scientific), 1X of non-essential amino acids (NAA; Life Technologies), 1X of Penicillin-Streptomycin (Life Technologies) were added to the flask. After 2 days of incubation, the cell culture supernatant was harvested and the percentage of FBS adjusted to 10%. Aliquots were stored at -80°C until used.”

• Lines 111-112 - “with the standard technique of forced salivation (21), wings and legs were removed from each mosquito, and the proboscis was inserted into a 20 μL tip containing 5 μL FBS. – The phrasing is a little confusing, as it sounds like wings/legs were removed for mosquitoes feeding on the filter paper too. Should state the time points post-exposure when salivation experiments were performed.

This missing information has been added in line 123: “mosquitoes were examined at 3 and 7 days after infection”

3) The introduction would be better structured if the first and second paragraphs were swapped.

We prefer to keep as it is: the first paragraph describes the mosquito physiology and its needs to get blood and sugar and the third paragraph states that we take advantage of these features to set up a surveillance system aiming at detecting in an anticipated manner any circulating viruses.

4) Lines 60-62 “Honey-coated cards have been used successfully to detect arbovirus circulation (namely, West Nile virus, Ross River virus, Barmah Forest virus, Japanese encephalitis virus and CHIKV) (16-19).” – what is different between this historical CHIKV study and the current study?

In Hall-Mendelin et al. (2010; PMID: 20534559), the design of experiments is different:

- Mosquitoes used are from a different region

- CHIKV strain was isolated from a patient in Australia and viral stocks were obtained after three passages on vertebrate (Vero) cells

- 12 days after experimental infection, mosquitoes were exposed to cards for 48 h

The authors found 75% (21/28) of mosquitoes that had expectorated the virus on honey-impregnated filters compared to 52% (14/27) of mosquitoes that have virus detected in saliva collected using a capillary tube. There was no significxant difference between the two methods.

5) Cards were homogenized in 500uL of medium. Saliva samples in 5uL FBS were added to 45uL medium. This is a 10x difference in media volume, and could have led to differences in detection rates between the two approaches, depending on their relative sensitivity. Was this accounted for in downstream calculations and statistical comparisons.

The dilution effect has been taken into account in our calculations and comparisons.

To quantify RNA copies in samples, signals were normalized to the standard curve using 10-fold serial dilutions of viral RNA (101 to 108). The number of RNA copies detected corresponded to the quantity of viral particles tested (see S1 Table).

6) The terms days post-exposure and days post-infection are both used throughout the manuscript. For the sake of consistency, only one should be used.

Dpe and dpi are two terms which are used in two different situations:

- dpe (days post-exposure ) refers to the numbers of days after exposure of mosquitoes to honey-impregnated cards

- dpi (days post-infection) corresponds to the number of days after depositing viral dilutions on cards

7) The Fig 2 legend does not do a good job of describing the figure and should be re-written. From the data in these panels, some treatment groups have a smaller sample size than anticipated based on what is described in the methods (one has an N of 3). The sample sizes for individual treatment groups should be described somewhere. Legends should also state whether data were pooled across replicates, or if they represent a single experiment.

We have added the following information in the legend of Fig. 2: “Each dot represents an individual mosquito. Between 11 to 60 mosquitoes were analyzed for Ae. aegypti and 5 to 46 for Ae. albopictus. Three replicates were performed per condition. Bars indicate the mean. ns (non-significant) indicates the lack of statistical significance (p > 0.05).”

8) The discussion section could be improved on in the following ways:

• The section overlooks the authors results which demonstrate that filter cards are less accurate at detecting virus in saliva than the forced salivation method. An objective discussion of the strengths and weaknesses of the two approaches is important.

• The EIP discussion from line 252 is very general and does not provide supporting evidence to the authors argument that honey-impregnated filter cards are a useful tool for arbovirus surveillance.

• Some key advantages of the filter cards are overlooked: they could be deployed in the field, left for a week and collected – a feeding station for surveillance, similar to sentinel birds for WNV. This would offer a direct measurement not feasible for other approaches. Honey promotes RNA stability, overcoming a major weakness with other detection methods.

The discussion section has been completely rearranged:

- the first paragraph describes the advantages of filter papers (non destructive, track viral dynamics in mosquitoes) and underlines the disadvantage of not being able to identify the vector species

- the second paragraph deals with the forced salivation technique used to measure the vector competence; the method is accurate but only examine mosquitoes which can be reared and infected in laboratory conditions.

9) Abstract – “Detection of virus in mosquito saliva using honey-impregnated filter papers seems to be a promising method.” – This sentence is vague. Why, specifically, is this a promising method?

As suggested, we have added details in the sentence: “Detection of virus in mosquito saliva using honey-impregnated filter papers seems to be a promising method as it is non-destructive and allows monitoring the viral excretion dynamics over time from the same mosquito.”

10) Lines 17-18 “Mosquito control is implemented when arboviruses are detected in patients or in field collected mosquitoes.” – It is more commonly implemented pre-emptively in areas of known transmission.

Usually, vector control is implemented when the virus is detected in patients or field-collected mosquitoes (it is at least what we do in France). The main concern is to limit the use of insecticides to avoid selecting insecticide-reisistance mosquitoes and chemical pollution.

11) Lines 162-163 - “and as the virus dilution increased (Kruskal-Wallis test: p <0.001).” – This reads as though more virus was detected when samples were more diluted, but the figure shows the opposite effect.

We found more virus on cards where we have deposited more virus and thus, where the dilution is the lowest. For a better understanding, we have rephrased as follows: “Moreover, the number of viral RNA detected on cards decreased as the virus dilution increased (Kruskal-Wallis test: p <0.001).” in lines 175-177.

12) Line 169 – “50% preserving better viral RNA.” This is confusing and needs to be rephrased.

We have replaced by “Altogether, the number of viral RNA detected on filter papers did not vary according to the dpi but vary with the honey dilution; the 50% dilution of honey preserved better viral RNA on cards.”

13) Lines 178-179 – “Detection of viral RNA on honey-impregnated filter papers proposed to CHIKV orally-infected mosquitoes” – This is confusing and needs to be rephrased.

We suggest: “Detection of viral RNA on honey-impregnated filter papers placed in contact with mosquitoes orally infected with a CHIKV-infectious blood meal”

14) Lines 180-183 “To define whether filter papers can replace saliva collection by forced salivation, Ae. aegypti and Ae. albopictus previously exposed to an infectious blood meal were put individually in contact with filter papers prepared under two conditions (status unchanged and changed) and examined at 3 and 7 dpe.” – This is a little difficult to understand, and it would be helpful if definitions for ‘changed’ and ‘unchanged’ were included in the text. In the legend for figure 2, the process is not well defined.

To be more accurate, we have added in lines 199-201: “the status “unchanged” refers to filter papers that were placed on the top of the tube and kept as it is until the day of examination and the status “changed” to filter papers placed on the top of the tube that were changed one day before the examination.”

15) Line 212 “viral RNA excreted” – should read viral RNA expectorated

It has been modified.

16) Lines 248-249 “This method has the advantage of being non-destructive as is the technique of forced salivation” – I think the phrasing here needs a little work as it makes it sound as though you are arguing that forced salivation is not destructive, even though it typically involves removing the legs and wings of the mosquitoes involved.

We have replaced by: “This method being non-destructive, monitoring the viral excretion dynamics over time is made possible from the same mosquito.”

17) Table 1 – mean transmission efficiency units (% I assume) should be included in the table.

% have been added.

Answers to Reviewer #2

1) The authors should clarify how their work is novel compared to the already published study that was using the same approach to detect CHIKV RNA in mosquito saliva (doi: 10.1073/pnas.1002040107).

In Hall-Mendelin et al. (2010; PMID: 20534559), authors (i) used Aedes aegypti and not Aedes albopictus, (ii) CHIKV orally-infected mosquitoes were placed in contact with honey-soaked cards at day 12 post-infection, (iii) cards were examined 2 days after. They found no significant difference between the proportion of mosquitoes that had expectorated the virus on honey-impregnated filters and the proportion of mosquitoes that have virus detected in saliva collected using a capillary tube.

Our study shows that filter papers can replace the forced salivation technique to detect viral RNA in saliva of Ae. aegypti and Ae. albopictus. CHIKV RNA can be detected in saliva deposited on filter papers until 7 days after oral infection of mosquitoes and RNA can be detected on filter papers coated with a 50% honey solution for up to 7 days after the virus has been deposited on the filter. It has been inserted in lines 250-253: “Moreover, viral CHIKV RNA can be detected 7 days after being deposited on filter papers impregnated with a 50% honey solution. To summarize, viral RNA can be detected on filter papers 14 days after oral infection of mosquitoes.”

2) As the transmission efficiency is mosquito dependent, it is rather complicated to compare the results obtained with the honey cards and the forced salivation without taking this parameter into account. Using forced salivation on the mosquitoes that were exposed to the honey cards to detect viral RNA would have allowed a better comparison between the different detection methods.

We agree with reviewer’s comment. The ideal experimental design would be to: (1) expose orally infected mosquitoes to filter papers and (2) immediately after to collect saliva using the forced salivation technique. The main problem is to make sure that the quantity virus expectorated by the mosquito using the method (1) will not interfere with the quantity obtained with the method (2).

3) The authors should explain the aim of testing the honey cards in two distinct conditions (changed and unchanged).

As suggested, we have inserted in lines 118-120: “To define whether the quantity of virus detected depends on contact time between cards and mosquitoes, cards were unchanged until day of examination or changed one day before examination.”

4) Related to the Table 1: The authors indicated that the replicates obtained for each condition were similar (line 185). Could the authors confirm this statement as the variations for A. albopictus at Day 3 vary from 0 to 71 (card unchanged) and from 20 to 64.28 (card changed).

The details of our statistical analysis are presented in the following table

 Aedes aegypti Aedes albopictus

 Pearson chi2 P Pearson chi2 P

Card unchanged Day 3 1.9922 0.369 7.5459 0.023

 Day 7 2.4257 0.297 1.4450 0.486

Card changed Day 3 1.8022 0.406 3.1484 0.207

 Day 7 2.8814 0.237 1.1625 0.559

Saliva collection Day 3 2.5000 0.287 3.6364 0.057

 Day 7 3.3190 0.190 10.4348 0.005

Two P values are above 0.05 ‘in bold) and as such, are considered significantly different. However, when taking into count the number of tests and adjusting the significance level to the number of tests, these P values are below the significance threshold.

P (Aedes aegypti) from max to min P (Aedes albopictus from max to min Number of tests run Threshold

0,406 0,559 1 0,05

0,36 0,486 2 0,025

0,297 0,207 3 0,0125

0,287 0,057 4 0,00625

0,237 0,023 5 0,003125

0,19 0,005 6 0,0015625

We have added inlines 160-161: “If necessary, the significance level of each test was adjusted based on the number of tests run, according to the sequential method of Bonferroni.”

5) The titer of the virus is indicated as 106.5 pfu/mL. This annotation is quite unusual, could the authors confirm that they mean 3.16 x 106 pfu/mL?

Yes we confirm.

6) The authors should provide, to the best extent possible, all the information in the figure legend and make table clearer so the reader can understand the figures:

a. In Figure 1, it is not clear what each dot represents. Indeed, if one dot represents one honey card then the authors should make visible on the graph all the data points, even when no viral RNA copies were detected. Another alternative would be to indicate above each virus dilution how many honey cards were tested.

We have added in the ms in the legend of Fig. 1 “Each dot represents one card; detailed information in S2 Table.”

An S2 Table has been added to the ms.

b. In Table 1, it is not clear what the unbracketed number refer to.

They are percentages; % have been added.

c. In Table 2, it is not clear how the treatments were spread across the two different mosquito species.

This table has become the S3 Table.

The proportions of mosquitoes showing transmission were compared between species, day post-exposure (dpe) and treatments using logistic regression models.

Dpe (3, 7) and treatments affected significantly TE while mosquito species did not. Within “treatments”, mosquitoes are considered together without distinction of species.

d. In Table 2, it is difficult to assess what data comparison the p-values are referring to.

There is no Table 2. This table has become the S3 Table.

We have rearranged the table.

e. In Figure 2, statistic could be indicated even if p-values are not significant.

In the legend of Figure 2, we have indicated: “ns (non-significant) indicates the lack of statistical significance (p > 0.05).”

7) Please refer to the figures in the text when necessary.

a. No mention of Figure 1a, 1b.

It has been corrected.

b. Text in lane 199 and between lane 199-203 should refer to table S2.

“When examining the number of viral RNA detected, loads were significantly higher at 7 dpe compared to 3 dpe (Student‘s test: p < 10-3) and did not vary according to the mosquito species (Student’s test: p = 0.42) and treatments (ANOVA: p = 0.75)” did not refer to S3 Table.

---

## [Decision Letter · Decision Letter 1]

16 Mar 2021

PONE-D-20-36874R1

A laboratory-based study to explore the use of honey‑impregnated cards to detect chikungunya virus in mosquito saliva

PLOS ONE

Dear Dr. Failloux,

Thank you for submitting your manuscript to PLOS ONE. After careful consideration, we feel that it has merit but does not fully meet PLOS ONE’s publication criteria as it currently stands. Therefore, we invite you to submit a revised version of the manuscript that addresses the points raised during the review process.

Both reviewers think you did a great job responding to their queries after the first round of review but there are still a few issues to be responded and/or incorporated into your manuscript before a decision can be made.

We look forward to receiving your revised manuscript.

Kind regards,

Luciano Andrade Moreira, PhD

Academic Editor

PLOS ONE

Journal Requirements:

Reviewers' comments:

Reviewer's Responses to Questions

**Comments to the Author**

1. If the authors have adequately addressed your comments raised in a previous round of review and you feel that this manuscript is now acceptable for publication, you may indicate that here to bypass the “Comments to the Author” section, enter your conflict of interest statement in the “Confidential to Editor” section, and submit your "Accept" recommendation.

Reviewer #1: (No Response)

Reviewer #2: (No Response)

2. Is the manuscript technically sound, and do the data support the conclusions?

Reviewer #1: Yes

Reviewer #2: Yes

3. Has the statistical analysis been performed appropriately and rigorously? 

Reviewer #1: Yes

Reviewer #2: Yes

4. Have the authors made all data underlying the findings in their manuscript fully available?

Reviewer #1: Yes

Reviewer #2: Yes

5. Is the manuscript presented in an intelligible fashion and written in standard English?

Reviewer #1: Yes

Reviewer #2: Yes

6. Review Comments to the Author

Reviewer #1: Most of my comments have been sufficiently addressed but two have not.

Initial question - #2a - Were any uninfected controls to help gauge the false positive rate?

Answer - To answer properly to this question, we need more details

Follow-up: I will clarify my comment: As a control in experimental infections, it’s worthwhile to include a mock infection treatment group - mosquitoes that were not exposed to CHIKV-infected blood, and were instead fed naïve blood, or blood plus sterile culture media. Mosquitoes from this treatment would then be allowed to salivate on the honey-impregnated cards in the same manner as CHIKV-exposed mosquitoes in order to assess the accuracy of the viral detection/quantification methodology. If a similar control group was included in the experiments, the results should be discussed in the paper. If no control group was included that should be specified in the methodology.

Initial question - #4 - Lines 60-62 “Honey-coated cards have been used successfully to detect arbovirus circulation (namely, West Nile virus, Ross River virus, Barmah Forest virus, Japanese encephalitis virus and CHIKV) (16-19).” – what is different between this historical CHIKV study and the current study?

In Hall-Mendelin et al. (2010; PMID: 20534559),

Answer - the design of experiments is different: -Mosquitoes used are from a different region

-CHIKV strain was isolated from a patient in Australia and viral stocks were obtained after three passages on vertebrate (Vero) cells

-12 days after experimental infection, mosquitoes were exposed to cards for 48 h

The authors found 75% (21/28) of mosquitoes that had expectorated the virus on honey-impregnated filters compared to 52% (14/27) of mosquitoes that have virus detected in saliva collected using a capillary tube. There was no significxant difference between the two methods.

Follow-up: These details need to be included somewhere in the introduction or discussion.

Reviewer #2: The authors have taken into consideration and addressed the majority of the comments provided in the first round of revision.

However, two minor comments remain to be fully addressed:

1) The authors should incorporate into the discussion few sentences to explain how they work complete the previous work from Hall-Mendelin et al., 2010 (doi: 10.1073/pnas.1002040107). This will help the readers to better understand the novelty of this paper.

2) While the authors have modified the table 1 to incorporate percentages, many of percent symboles are still missing. Could the authors carefully correct this point?

7. PLOS authors have the option to publish the peer review history of their article (what does this mean?). If published, this will include your full peer review and any attached files.

Reviewer #1: No

Reviewer #2: **Yes: **Benjamin Voisin

---

## [Author Response · Author response to Decision Letter 1]

17 Mar 2021

Answers to Reviewer #1:

1) Initial question - #2a - Were any uninfected controls to help gauge the false positive rate?

Answer - To answer properly to this question, we need more details

Follow-up: I will clarify my comment: As a control in experimental infections, it’s worthwhile to include a mock infection treatment group - mosquitoes that were not exposed to CHIKV-infected blood, and were instead fed naïve blood, or blood plus sterile culture media. Mosquitoes from this treatment would then be allowed to salivate on the honey-impregnated cards in the same manner as CHIKV-exposed mosquitoes in order to assess the accuracy of the viral detection/quantification methodology. If a similar control group was included in the experiments, the results should be discussed in the paper. If no control group was included that should be specified in the methodology.

We understand but we did not expose mosquitoes to a non-infectious blood meal.

We have inserted this statement in line 128: “Non-infected mosquitoes (exposed to non-infectious blood or culture media) were not tested”.

1) Initial question - #4 - Lines 60-62 “Honey-coated cards have been used successfully to detect arbovirus circulation (namely, West Nile virus, Ross River virus, Barmah Forest virus, Japanese encephalitis virus and CHIKV) (16-19).” – what is different between this historical CHIKV study and the current study?

In Hall-Mendelin et al. (2010; PMID: 20534559),

Answer

- the design of experiments is different:

-Mosquitoes used are from a different region

-CHIKV strain was isolated from a patient in Australia and viral stocks were obtained after three passages on vertebrate (Vero) cells

-12 days after experimental infection, mosquitoes were exposed to cards for 48 h

The authors found 75% (21/28) of mosquitoes that had expectorated the virus on honey-impregnated filters compared to 52% (14/27) of mosquitoes that have virus detected in saliva collected using a capillary tube. There was no significxant difference between the two methods.

Follow-up: These details need to be included somewhere in the introduction or discussion.

As suggested, we have added in lines 261-263: “Its gives results comparable to those obtained using the standard forced salivation technique of mosquitoes (16)”.

Answers to Reviewer #2

1) The authors should incorporate into the discussion few sentences to explain how they work complete the previous work from Hall-Mendelin et al., 2010 (doi: 10.1073/pnas.1002040107). This will help the readers to better understand the novelty of this paper.

We have added in lines 261-263: “Its gives results comparable to those obtained using the standard forced salivation technique of mosquitoes (16)”.

2) While the authors have modified the table 1 to incorporate percentages, many of percent symboles are still missing. Could the authors carefully correct this point?

Thanks for noticing it. It has been corrected.

---

## [Editor Report · Decision Letter 2]

19 Mar 2021

A laboratory-based study to explore the use of honey‑impregnated cards to detect chikungunya virus in mosquito saliva

PONE-D-20-36874R2

Dear Dr. Failloux,

We’re pleased to inform you that your manuscript has been judged scientifically suitable for publication and will be formally accepted for publication once it meets all outstanding technical requirements.

Kind regards,

Luciano Andrade Moreira, PhD

Academic Editor

PLOS ONE
---

## [Editor Report · Acceptance letter]

23 Mar 2021

PONE-D-20-36874R2 

A laboratory-based study to explore the use of honey‑impregnated cards to detect chikungunya virus in mosquito saliva 

Dear Dr. Failloux:

I'm pleased to inform you that your manuscript has been deemed suitable for publication in PLOS ONE. Congratulations! Your manuscript is now with our production department. 

Kind regards, 

on behalf of

Dr. Luciano Andrade Moreira 

Academic Editor

PLOS ONE